# D-MFPN: A Doppler Feature Matrix Fused with a Multilayer Feature Pyramid Network for SAR Ship Detection

Yucheng Zhou [1,2,3], Kun Fu [1,2,3], Bing Han [1,2,3,*], Junxin Yang [1,2,3], Zongxu Pan [1,2,3], Yuxin Hu [1,2,3] and Di Yin [1,2,3]

1    Aerospace Information Research Institute, Chinese Academy of Sciences, Beijing 100094, China
2    Key Laboratory of Technology in Geo-Spatial Information Processing and Application System, Chinese Academy of Sciences, Beijing 100190, China
3    School of Electronic, Electrical and Communication Engineering, University of Chinese Academy of Sciences, Beijing 101408, China
*    Correspondence: han_bing@mail.ie.ac.cn; Tel.: +86-010-5888-7208 (ext. 8956)

**Abstract:** Ship detection from synthetic aperture radar (SAR) images has become a major research field in recent years. It plays a major role in monitoring the ocean, marine rescue activities, and marine safety warnings. However, there are still some factors that restrict further improvements in detecting performance, e.g., multi-scale ship transformation and unfocused images caused by motion. In order to resolve these issues, in this paper, a doppler feature matrix fused with a multi-layer feature pyramid network (D-MFPN) is proposed for SAR ship detection. The D-MFPN takes single-look complex image data as input and consists of two branches: the image branch designs a multi-layer feature pyramid network to enhance the positioning capacity for large ships combined with an attention module to refine the feature map's expressiveness, and the doppler branch aims to build a feature matrix that characterizes the ship's motion state by estimating the doppler center frequency and frequency modulation rate offset. To confirm the validity of each branch, individual ablation experiments are conducted. The experimental results on the Gaofen-3 satellite ship dataset illustrate the D-MFPN's optimal performance in defocused ship detection tasks compared with six other competitive convolutional neural network (CNN)-based SAR ship detectors. Its satisfactory results demonstrate the application value of the deep-learning model fused with doppler features in the field of SAR ship detection.

**Keywords:** ship detection; synthetic aperture radar (SAR)

## 1. Introduction

As a high-resolution microwave imaging radar, synthetic aperture radar (SAR) has unique strengths in various planetary observation tasks, such as its excellent all-day and all-weather working capacity, especially in marine observation [1]. As a part of marine missions, SAR ship detection is of great value in marine monitoring [2–6]. Thus, the detection basis of ships using SAR has become a focus of marine research.

The pixel intensity information of an SAR image is related to the target's scattering cross-section. Oceans often appear dark black in images, while ships are bright white. However, offshore islands and background noise will also appear in a similar bright white, and the defocusing produced by a ship moving at speed when an SAR image is taken will also cause serious geometric distortion, showing an irregular shape. Therefore, it is a challenging task to accurately distinguish and precisely locate ships from other targets in SAR images under real conditions. Traditional SAR ship target detection methods such as CFAR [7,8], and template matching-based methods [9,10] are generally divided into three steps: preprocessing, manual feature extraction, and setting thresholds or classifiers for detection to obtain the final result [11–13]. CFAR establishes the best decision by estimating the noise threshold and information, such as the statistical characteristics of signal and noise. However, the probability of a false alarm in the clutter edge area is higher than the

center area. The algorithm based on template matching distinguishes ships according to the size, area, and texture features, which need to be matched according to the template library established by experts. Template matching algorithms will always be affected by the background statistical area. In general, traditional SAR ship detectors generally use complex algorithms, have weak transfer ability, and have cumbersome manual features. In addition, due to the backscatter imaging mechanism, the traditional algorithm is highly sensitive to the geometric features of the target extracted from the SAR image, and the defocusing caused by the motion of the ship relative to the radar platform will dramatically reduce the performance of the algorithm.

At present, with the continuous development of neural network theory [14], SAR ship detection based on deep-learning models has become a current mainstream method [15–18]. For instance, Li et al. [19] proposed a new dataset and four strategies to improve detection performance based on the Faster RCNN [20] algorithm. Lin et al. [21] designed a squeeze and excitation rank mechanism to improve detection performance. Based on You Only Look Once v4 (YOLOv4) [22], Jiang et al. [23] integrated a multi-channel-fusion SAR image processing method that makes full use of image information and the network's ability to extract features and refined the network for three-channel images to compensate for the loss of accuracy. Based on the Swin Transformer [24], Li et al. [25] adopted a feature enhancement Swin transformer (FESwin) and an adjacent feature fusion (AFF) module to boost performance. Based on RetinaNet [26], Gao et al. [27] proposed a polarization-feature-driven neural network for compact polarimetric (CP) SAR data. Based on FCOS [28], Zhu et al. [29] redesigned the feature extraction module.

CNN-based algorithms have better feature extraction and generalization capabilities than traditional detectors. However, due to the differences in scale between ships, small targets lack detailed information on a scaled feature map and cause false alarms. Therefore, designing a detection model for small targets that adapt to multi-scale and complex backgrounds has become the main research direction in the field of SAR ship detection in recent years. The feature pyramid network (FPN) [30] has been the best method to solve multi-objective problems since it was proposed by Lin et al. For different incident angles, resolutions, satellites, etc., SAR ships possess various sizes. FPN uses feature maps of different scales to extract information at different levels of the picture and uses multi-scale feature information fusion to enhance the expression ability of ship targets, which enables better performance. For the ship detection problem, many scholars have made improvements on the basis of FPN. Zhu et al. [31] combined DB-FPN with YOLO to improve overall detection capability. Cui et al. [15] adopted an attention-guided balanced feature pyramid network (A-BFPN) to better exploit semantic and multi-level complementary features.

Compared to ordinary optical images, SAR echo signals contain complex features. Deep-learning SAR ship inspection models mostly use the amplitude information of SAR image as input, and the underutilization of complex information restricts the upper limits of these models. In order to further improve the utilization of the characteristics of SAR data, Xiang et al. [32] proposed a ship detection method in range-compressed SAR data that employs complex signal kurtosis (CSK) to prescreen potential ship areas and then apply a convolutional neural network (CNN) based discrimination to obtain potential ship areas. The results show that the algorithm works well on range-compressed SAR data. Zhang et al. [33] proposed a Polarization Fusion Network with Geometric Feature Embedding (PFGFE-NET) to comprehensively scan SAR ships using VV and VH polarized pictures and proposed a method to describe SAR ships between different polarization modes. Zhang et al. [34] designed a complex-valued CNN network (CV-CNN) specially used for SAR image interpretation using the amplitude and phase information of SLC data. All elements in the CNN, including input and output layers, volume convolution layers, activation functions, and pooling layers, are all extended to the complex domain, but these methods are only applicable to PolSAR images. In order to utilize the target feature information contained in the monopolar SAR phase, Huang et al. [35] introduced a deep-learning framework

dedicated to complex-valued radar images. Using CNN to extract spatial features from intensity images and ResNet-18 as the pre-training model, they generated feature maps with scales of 64, 128, 256, and 512 to learn hierarchical features at different scales. The joint time-frequency analysis method was used to learn the physical characteristics of the target in the frequency domain, reveal the relationship between the backscatter diversity of the ground target and the range and azimuth frequency, and align the target for a final prediction along with spatial texture features.

At present, complex value-based networks are mostly used in the classification task of open-source ship datasets, such as Opensarship. The selected ship slices often have obvious geometric characteristics, and the position of the ship slices can be distinguished easily. However, for the ship detection task, ships moving at high speeds are more likely to lose geometric features in the SAR imaging process, so it is therefore necessary to employ more abundant auxiliary features to help with target recognition. So far, few works have attempted to solve the moving ship detection task using doppler features. Therefore, a novel doppler feature matrix fused with a multi-layer feature pyramid network is proposed in this paper. Our work focuses on three areas. The first is the establishment of the dataset. To extract doppler features, we detail a single-look complex image ship dataset composed of Gaofen-3 satellite image ship slices. Secondly, we propose a detection framework that fuses doppler features. We extract the doppler center frequency and frequency modulation rate offset to characterize the ship's motion state from the doppler domain, and construct the doppler characteristic matrix. We then fuse it with spatial features to carry out a ship inspection. Finally, there is an improvement in the network structure for the image branch, and a bottom-up pyramid structure is designed to transmit the position information for large ships. This combines external attention (EA) and coordinate attention (CA) modules to solve the multi-scale moving ship detection problem.

Based on the above three points, we propose an SAR ship detection pipeline and conduct an experiment on the complex SAR ship dataset. The experimental results show that, compared with other deep-learning target detection algorithms, the D-MFPN can obtain the best performance in both inshore and offshore scenarios.

The main contributions of this paper are as follows:

- An SAR single-look complex ship dataset is constructed.
- We design a network structure based on FPN, using a bottom-up pyramid to transfer position information and refining features by combining CA and EA modules.
- An SAR ship detection framework that integrates doppler features and spatial features is proposed, combining them to improve the performance of the model.

The rest of the paper is organized as follows: Section 2 presents the entire detection model. Section 3 presents the the dataset used, experiments, and results.

## 2. D-MFPN

The D-MFPN consists of two branches. The image branch designs a multi-layer feature pyramid network to enhance the positioning capacity for large ships. The doppler branch aims to build a feature matrix that characterizes a ship's motion state by estimating the doppler center frequency and the frequency modulation rate offset. The overall structure of the D-MFPN is shown in Figure 1.

In this section, the implementation steps of the D-MFPN for SAR ship detection are described. Firstly, the motivation and overall structure of the multi-layer feature pyramid, which is designed for magnitude images is described. The method of using the complex matrix to obtain the doppler feature matrix is also given. At the end of this section, the method of combining the two is introduced in detail.

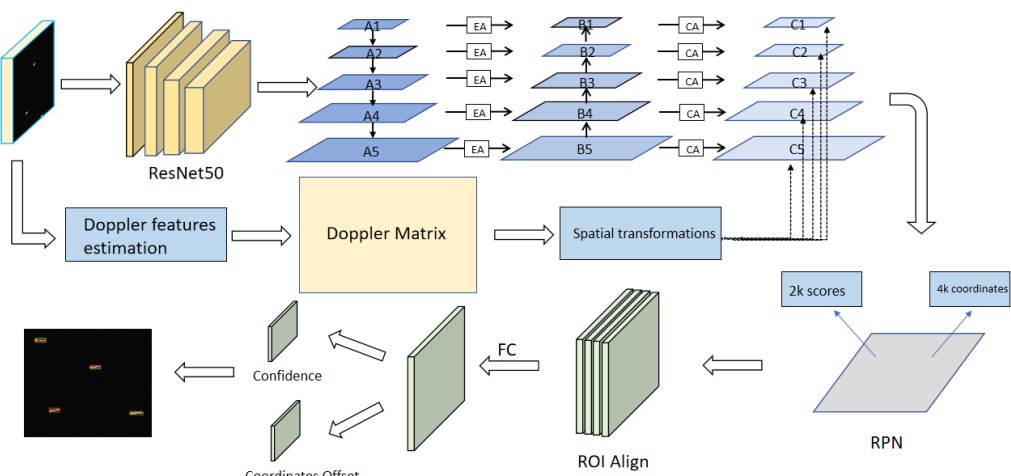

**Figure 1.** The architecture of the D-MFPN.

### 2.1. Multi-Layer Feature Pyramid Network

For the SAR ship detection task, small-scale targets, multi-scale feature imbalance, and complex background interference are all important factors that hinder the accuracy of the model. The CNN-based convolutional neural network is a composite structure composed of multiple convolutional layers. With the continuous increase in the network depth, high-level feature representations can be learned, but the small ship characteristics will be weakened with the increasing receptive field, leading to missed detections. In addition, the deep network structure will generate multiple nonlinear activation function partial derivatives or continuous multiplication of weight parameters during back propagation, which can easily cause gradient disappearance or gradient explosion. For small objects, the pyramid structure adopted by the FPN uses feature maps of different scales to be sent to the detection head, which alleviates the multi-scale problem to a certain extent, although its feature weakening remains unresolved. The top-down structure causes the direction of semantic information of the top layer to flow downward.However, the position information of the bottom layer is not propagated to the top layer, which also leads to inaccurate positioning of large ships.

To solve the above problems, the D-MFPN uses resnet-50 as a backbone. The CNN structure independently extracts the features of the labeled data, effectively avoiding the traditional complex feature design. The residual block module also prevents overfitting and vanishing gradient problems, greatly increasing the network depth. The residual structure of resnet50 consists of multiple structures called a "bottlenet" (as shown in Figure 2), and finally, five feature maps of different scales (conv1, conv2, conv3, conv4, conv5) are obtained.

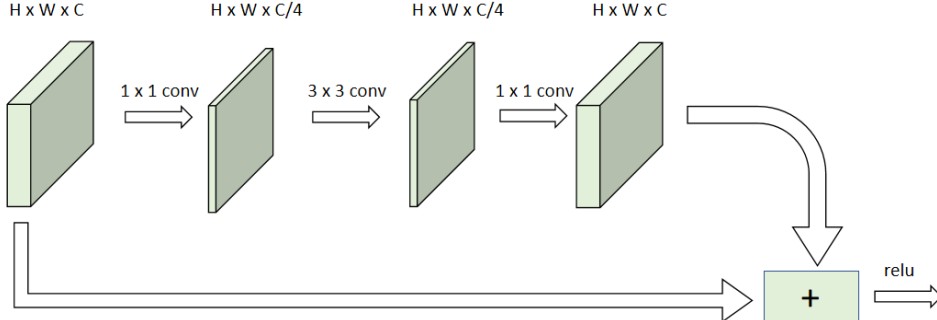

**Figure 2.** The architecture of a residual bottlenet.

With the continuous increase in the convolution stride and the number of residual network layers, the spatial resolution of the feature map gradually decreases, and the number of channels increases accordingly. The underlying feature maps contain accurate strong location information but lack semantic information. Therefore, the FPN uses the information to obtain the final combined features from different layers in the CNN network. As shown in Figure 3, the FPN adopts horizontal connections and adds top-down paths to improve the expressiveness of the entire pyramid. However, low-level location information from the bottom of the pyramid is not transferred to the top. This directly leads to the wrong localization of the bounding boxes of large ships, thus degrading the performance. To solve this problem, the D-MFPN designs a bottom-up pyramid structure to enhance information flow. The location information of the bottom layer is passed upwards using a bottom-up pyramid structure. In addition to strengthening the positioning effect of large ships, in order to further improve the expression ability of feature maps of each layer, we also use two attention modules to enhance the association between each pixel and channel.

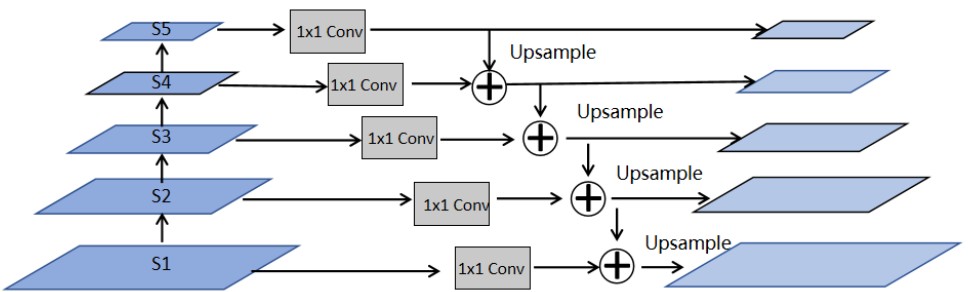

**Figure 3.** The structure of the feature pyramid network (FPN).

The attention methods are inspired by the human visual attention mechanism, the essence of which is to let the model ignore irrelevant information through a series of related calculations, pay more attention to the key information we want it to focus on, and obtain long-distance correlations between pixels. Usually, the way to obtain the attention weight is to imagine the constituent elements in the source as a series of key and value data pairs, and for an element query in the target, calculate the similarity between the query and each key or correlation, obtain the weight coefficient of the value corresponding to each key, and then weight and sum the value to obtain the final attention value. In the ship detection task, the background of the far sea is relatively monotonous. Different port backgrounds near the coast can also have a certain degree of similarity. Therefore, if an attention mechanism can be found to learn the implicit relationship between different samples, it will greatly improve the detection performance. Considering the above perspectives, we use the EA module, which uses two linear layers to replace the inputs that make the network learnable from the obtained keys and values. A normalization layer is used to obtain the similarity matrix between query and key. A learnable weight matrix is used to replace the key and value obtained by linear transformation according to the input itself so that the EA module can update the parameters according to the entire training set to better fit the data characteristics. At the same time, EA has linear complexity, which implicitly considers the relationship between different samples.

Figure 4 shows the implementation process of EA. In order to also make the attention mechanism consider the influence from other samples, EA designs two feature layers shared by all samples, $M_k$ and $M_v$. First, in the same way as the self-attention mechanism, the input feature map F is flattened as a query and then multiplied with the $W_k$ matrix using softmax normalization to obtain the correlation matrix between query and key. Finally, the same operation is performed with $M_v$ to obtain the output. The formula for this process is as follows:

$$A = (\alpha)_{i,j} = Norm(FM_k^T) \tag{1}$$

$$F_{out} = AM_v \tag{2}$$

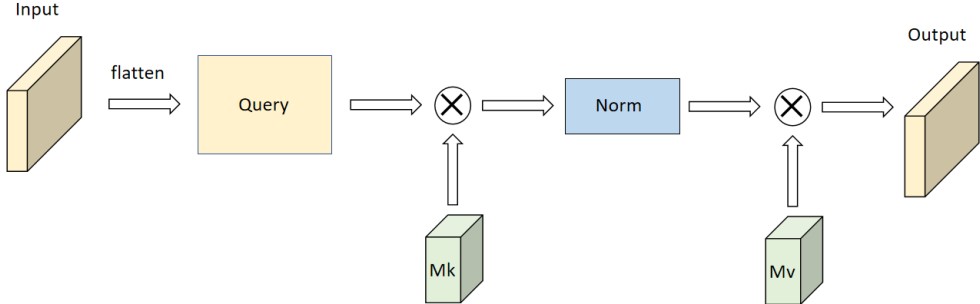

**Figure 4.** External attention. F represents the feature map flattened by the input, and A represents the attention feature map obtained from prior knowledge.

As shown in Figure 5, the location information of the bottom of the pyramid is transferred to the top (B5→B4→B3→B2→B1). In this way, the high-level feature map will contain more location information, which will improve the feature expression ability of large ships to a certain extent. In addition, before downsampling, the low-level feature maps are refined by EA, which improves the effect of upsampling.

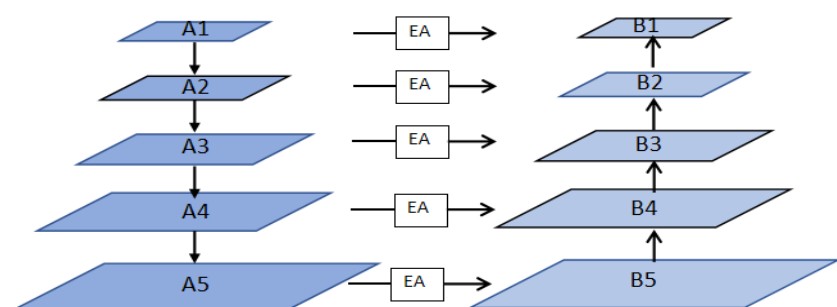

**Figure 5.** Structure of multi-layer feature pyramid.

After the feature map is obtained, it is fused with the underlying information. In order to further refine the location features of each layer in the feature pyramid and solve the problem of the low recognition rate of overlapping ships of different scales, the D-MFPN adds the CA module at this step. Research on attention mechanisms shows that inter-channel attention has a significant effect on improving model performance, but it often ignores the location information between pixels. As shown in Figure 6. CA encodes the relationship between channels and long-range location information, and the overall structure is divided into two steps: coordinated information embedding and coordinated attention generation.

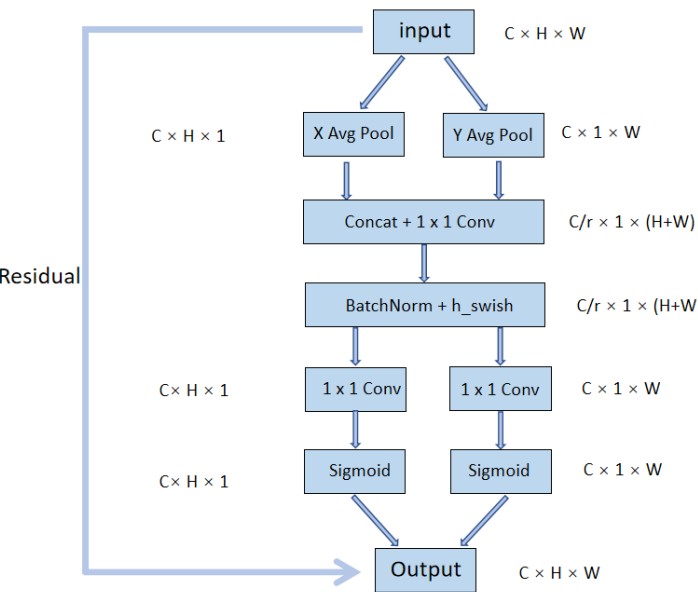

**Figure 6.** The structure of coordinate attention.

Channel attention often uses global pooling to encode spatial attention information, but it is often difficult to preserve the position information between pixels. The location information is crucial for the detection task to capture the local spatial structure. Therefore, coordinate information embedding decomposes the global pooling into average pooling along the direction of height H and width W of the input feature map. Encoded along the horizontal and vertical coordinates, it can be expressed as:

$$z_c^h(h) = \frac{1}{W} \sum_{0 \leq i < W} x_c(h, i)$$

$$z_c^w(w) = \frac{1}{H} \sum_{0 \leq j < H} x_c(j, w) \tag{3}$$

The difference between aggregating features along the horizontal and vertical directions separately and using global pooling directly is that these two transformations can help to obtain distance dependencies along one direction at the same time. Along the other direction, accurate location information can be preserved.

In order to make full use of the positional features obtained by coordinate information embedding, the coordinated attention generation step concatenates the two and then passes the $1 \times 1$ convolution function $F$

$$f = \delta(F([z^h, z^w])) \tag{4}$$

where $[\cdot, \cdot]$ represents the concatenation along spatial dimension; $\delta$ denotes the h_swish activation function; the shape of output $f$ is $(C/r, (H + W))$; and $r$ is the reduction ratio for controlling the block size as in the SE block. Then, we decompose f into components in the $h$ and $w$ directions and go through another $1 \times 1$ convolutional layer to turn it into a tensor with the same number of channels as the input X:

$$g^h = \sigma(F_h(f^h))$$

$$g^w = \sigma(F_w(f^w)) \tag{5}$$

where $F_h$ and $F_w$ are tensors with the same channels obtained with the decomposed vector after $1 \times 1$ convolution, $\sigma$ is the sigmoid function, and $g_w$ and $g_h$ are the final attention weights. The output of the CA module can be expressed as:

$$y_c(i, j) = x_c(i, j) \times g_c^h(i, j) \times g_c^w(i, j) \tag{6}$$

*2.2. Doppler Domain Feature Estimation*

In the process of processing SAR detection echo data, due to the movement of the target, the doppler modulation frequency rate and center frequency of the signal will shift, and the deviation of the modulation frequency will result in a defocused image. Conversely, the feature matrix composed of offsets can be used to describe the motion characteristics of the target. Therefore, by estimating the doppler feature offset of the echo signal, the information that characterizes the motion state of the target can be obtained.

The geometric relationship between the satellite and target moving in the direction of $\mathbf{V_m}$ is shown in Figure 7, where T represents the target point. According to the spaceborne SAR echo theory, the distance history of the stationary target T can be obtained as:

$$R(t) = R + \frac{\mathbf{V_{st}} \cdot \mathbf{R}}{R} t + \frac{1}{2} \left[ \frac{\mathbf{V_{st}} \cdot \mathbf{V_{st}}}{R} + \frac{\mathbf{A_{st}} \cdot \mathbf{R}}{R} - \frac{(\mathbf{V_{st}} \cdot \mathbf{R})^2}{R^3} \right] t^2 \tag{7}$$

In which

$$\begin{aligned}
\mathbf{R} &= \mathbf{R_s} - \mathbf{R_t} \\
\mathbf{V_{st}} &= \mathbf{V_s} - \mathbf{V_t} \\
\mathbf{A_{st}} &= \mathbf{A_s} - \mathbf{A_t}
\end{aligned} \tag{8}$$

where $\mathbf{R_s}$, $\mathbf{V_s}$, and $\mathbf{A_s}$ represent the satellite position, velocity, and acceleration vectors at the beam center time, respectively; $\mathbf{R_t}$, $\mathbf{V_t}$, and $\mathbf{A_t}$ represent the beam center time target position, velocity, and acceleration vectors, respectively; and $\mathbf{R}$, $\mathbf{V_{st}}$, and $\mathbf{A_{st}}$ represent the relative satellite position, velocity, and acceleration vector, respectively. If target T moves in a straight line at a uniform speed along the vector $\mathbf{V_m}$, the distance between the moving target and the radar is:

$$R'(\eta) = R + \frac{(\mathbf{V_{st}} + \mathbf{V_m}) \cdot \mathbf{R}}{R} \eta + \frac{1}{2} \left[ \frac{(\mathbf{V_{st}} + \mathbf{V_m}) \cdot (\mathbf{V_{st}} + \mathbf{V_m})}{R} + \frac{\mathbf{A_{st}} \cdot \mathbf{R}}{R} - \frac{((\mathbf{V_{st}} + \mathbf{V_m}) \cdot \mathbf{R})^2}{R} \right] \eta^2 \tag{9}$$

The instantaneous slant range error caused by the moving target is:

$$\begin{aligned}
\triangle R(\eta) = R'(\eta) - R(\eta) &\approx |\mathbf{V_y}| sin\gamma \cdot \eta \\
&+ \frac{1}{2} \left[ \frac{2|\mathbf{V_{st}}||\mathbf{V_x}| + |\mathbf{V_x}|^2 + |\mathbf{V_y}|^2 cos^2\gamma}{R} \right] \cdot \eta^2
\end{aligned} \tag{10}$$

where $\gamma$ is the incident angle, and the phase error changes the doppler information of the azimuth signal. According to the expression of the slant range error, the error of the azimuth doppler frequency can be described as:

$$\triangle f_d = \frac{2}{\lambda} \frac{d \triangle R(\eta)}{d\eta} = \frac{2}{\lambda} \left( |\mathbf{V_y}| sin\gamma + \left[ \frac{2|V_{st}||V_x| + |V_x|^2 + |V_y|^2 cos^2\gamma}{R} \right] \cdot \eta \right) \tag{11}$$

From the above formula, the changes in the doppler center frequency and doppler modulation frequency rate can be shown as:

$$
\begin{aligned}
\triangle f_{dc} &= \frac{2sin\gamma}{\lambda}|\mathbf{V_y}| \\
\triangle f_\gamma &= \frac{2}{\lambda R}\left(2|\mathbf{V_{st}}||\mathbf{V_x}| + |\mathbf{V_x}|^2 + |\mathbf{V_y}|^2 cos^2\gamma\right)
\end{aligned}
\tag{12}
$$

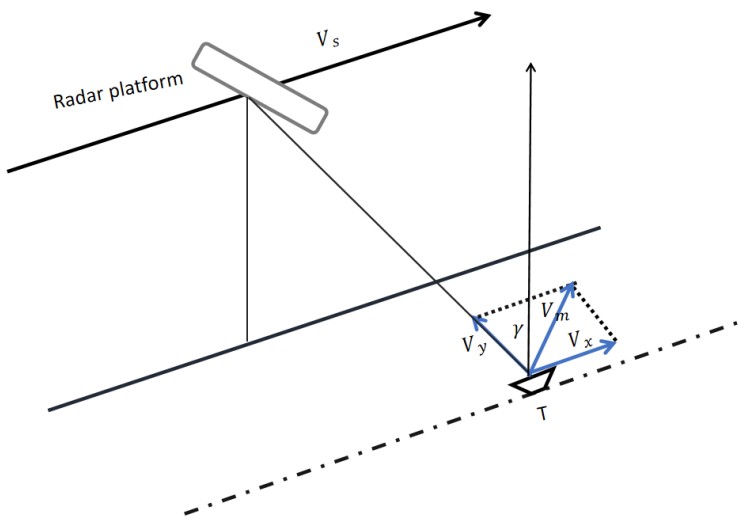

**Figure 7.** Schematic diagram of the geometric relationship between the satellite and the Earth.

From Equation (12), it can be seen that the doppler-center shift in the azimuth direction is related to the radial velocity of the target. In the D-MFPN, the processing flow of doppler migration for a complex image matrix is as follows. First, slice the data and perform the azimuth Fourier transform. Then, obtain the azimuth spectrum through fftshift. Since the characteristics represented by a single spectrum are not obvious, we adopt the range block method to incoherently stack the azimuth spectrum. The fitted parameters are given in Section 3. In order to eliminate the influence of outliers, a high-order fitting on the spectral features is conducted to obtain the doppler center shift features and finally make the doppler feature matrix the same size as the original image. In the D-MFPN, the input doppler center frequency shift matrix will first undergo a normalization process to map the eigenvalues between 0 and 1.

The shift in doppler center frequency can be used to describe the radial velocity of the ship relative to the radar platform, while the frequency shift due to the lateral velocity resulting in defocusing can be estimated using the mapdrift algorithm. Suppose the ideal azimuth signal is:

$$
s_a(\eta) = exp(j\pi K_a\eta^2) + j2\pi f_{dc}\eta),\ -\frac{T_s}{2} \le \eta \le \frac{T_s}{2}
\tag{13}
$$

where $\eta$ denotes the azimuth time, $K_a$ denotes the doppler modulation frequency, $f_{dc}$ denotes the doppler center, and $T_s$ denotes the synthetic aperture time. The azimuth signal with quadratic phase error is:

$$
g_a(\eta) = s_a(\eta)\cdot exp(jK_e\eta^2)
\tag{14}
$$

The mapdrift algorithm divides the entire synthetic aperture time into two sub-apertures that do not coincide with each other, as follows:

$$g_{a1}(\eta) = g_a(\eta - \frac{T_s}{4}) = s_a(\eta - \frac{T_s}{4}) \cdot exp(j(K_e\eta^2 - \frac{1}{2}K_sT_s\eta + \frac{1}{16}K_eT_s^2))$$
$$g_{a2}(\eta) = s_a(\eta + \frac{T_s}{4}) \cdot exp(j(K_e\eta^2 + \frac{1}{2}K_sT_s\eta + \frac{1}{16}K_eT_s^2))$$

(15)

In the above formula, $-\frac{T_s}{4} \leq \eta \leq \frac{T_s}{4}$. Neglecting constant quantities that are insignificant to focus quality, the second-order components of the quadratic phase error within the two sub-apertures are exactly the same, and the first-order components are the same in magnitude but opposite in sign. Among them, the effect of the second-order component on the matched filter is defocusing, and the effect of the first-order component on the matched filter is to change the peak position. When pulse-compressing both sub-apertures with the same matched filter, the signals of both sub-apertures are defocused, but the peaks are shifted at different positions. The deviation in the two peak positions can be calculated by cross-correlating the amplitudes of the compressed signals of the two sub-apertures.

The quadratic phase error of the chirp signal is the main cause of defocusing, which is mapped to the motion relationship and corresponds to the lateral motion of the target relative to the radar platform. However, the moving direction of the sea surface target is not completely perpendicular to the radar platform, and there are often both radial and lateral components. Therefore, the D-MFPN simultaneously estimates the offset of both as a feature to characterize the motion state.

### 2.3. Feature Fusion

Feature maps will take on multi-scale shapes after feature extraction from the magnitude image. Each point in the feature map maps the receptive field of different regions in the original image. After obtaining the doppler domain feature, they need to be spatially aligned with the points in the feature map. The magnitude image needs to transfer the convolutional and pooled spatial transformation information to the doppler domain, and average pooling can then be performed on the resulting doppler information matrix to ensure it has the same receptive field as the feature map. In order to preserve the ship's motion state information contained in the doppler matrix, during the feature fusion process, the D-MFPN does not perform additional convolution operations on the doppler branch after spatial alignment but directly concatenates space and doppler features as the input of the detection head. The entire feature fusion and spatial alignment module process is shown in Figure 8:

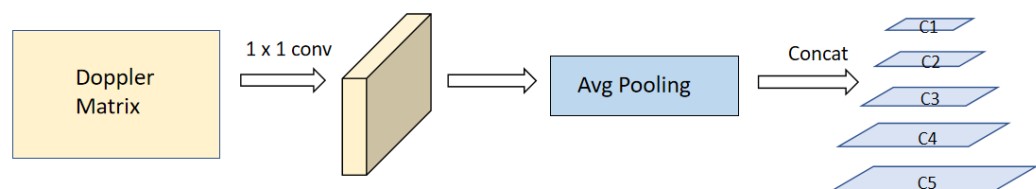

**Figure 8.** Feature fuse.

The fusion module firstly aligns the doppler domain feature matrix and its channel number with the last pyramid channel number using a $1 \times 1$ convolution. Then, for different feature map sizes of each layer, average pooling is used to spatially align them.

### 2.4. Loss Function

The D-MFPN is based on the Faster RCNN, so the loss function is similar to most two-stage object detection models. It is divided into classification loss and regression loss for optimization:

$$L(p_i, t_i) = \frac{1}{N_{cls}} \sum_i L_{cls}(p_i, p_i^*) + \lambda \frac{1}{N_{reg}} \sum_i p_i^* L_{reg}(t_i, t_i^*) \tag{16}$$

where $p_i$ indicates the probability that the i-th anchor is predicted to be the true label, $p_i^*$ is 1 for positive samples and 0 for negative samples, $t_i$ represents the parameter of the ith regression box, $t_i^*$ represents the GT box of ith anchor, $N_{cls}$ denotes the number of mini-batches, $N_{reg}$ denotes the number of anchor positions, and $L_{cls}$ represents the cross-entropy (CE) loss

$$L_{cls} = -\frac{1}{N} \sum_{i=1}^{N} p_i log(p_i^*) + (1 - p_i) log(1 - p_i^*) \tag{17}$$

using smooth L1 as regression loss $L_{reg}$. Compared with L1 loss, smooth L1 loss improves the problem of zero-point unsmoothing. Compared with L2 loss, it is not as sensitive to outliers when the value is large. The gradient changes are relatively smaller, and the training process is more stable.

$$L_{reg}(t_i, t_i^*) = smooth_{L1}(t_i - t_i^*)$$
$$smooth_{L1}(x) = \begin{cases} 0.5x^2 & if |x| < 1 \\ |x| - 0.5 & otherwise \end{cases} \tag{18}$$

## 3. Results

### 3.1. DATASET

Ship detection in synthetic aperture radar images has become a research hotspot for scholars, and many SAR ship datasets have been published in recent years, such as SSDD [19], LS-SSDD [36], AIR-SARShip1.0, HRSID, etc. However, the images of these datasets are often processed into jpg or png formats when they are published, which makes the model unable to fully utilize their doppler features. Therefore, in this paper, an SAR single-look complex image moving ship dataset is proposed by using Gaofen-3 satellite data to explore the problem of moving ship detection in the ocean (shown in Figure 9). The dataset consists of 31 large-scale images of 150,000 × 30,000 pixels. We selected 943 ship slices with a scale of 512 × 1024. The imaging modes consist of Strip-Map (UFS) and Fine Strip-Map1 (FSI), and the corresponding resolutions are 3 m and 10 m. All slices in the dataset are processed to mat format.

Defocusing in images of ships can severely distort the ship's geometry, making it easy to be mistaken for islands or background noise. In this case, it is difficult to make correct annotations, and the essence of defocusing is that for the chirp signal emitted by the SAR, a one-order shift is generated in the frequency domain, resulting in blurring during imaging. Therefore, in order to obtain slices of moving ships while ensuring the accuracy of annotation, the ship targets with obvious geometric features are first annotated. Due to the radial movement of the target along the radar platform, the doppler center frequency will be shifted, which will not cause defocusing. Lateral movement will cause deviations in the frequency modulation rate, resulting in defocusing. Most targets have both radial and lateral motion components. Therefore, as shown in Figure 10, we first perform a Fourier transform along the azimuthal direction and then add random primary and secondary phase offsets. Finally, we perform an inverse Fourier transform to obtain defocused image samples, which ensures the accuracy of the labeling.

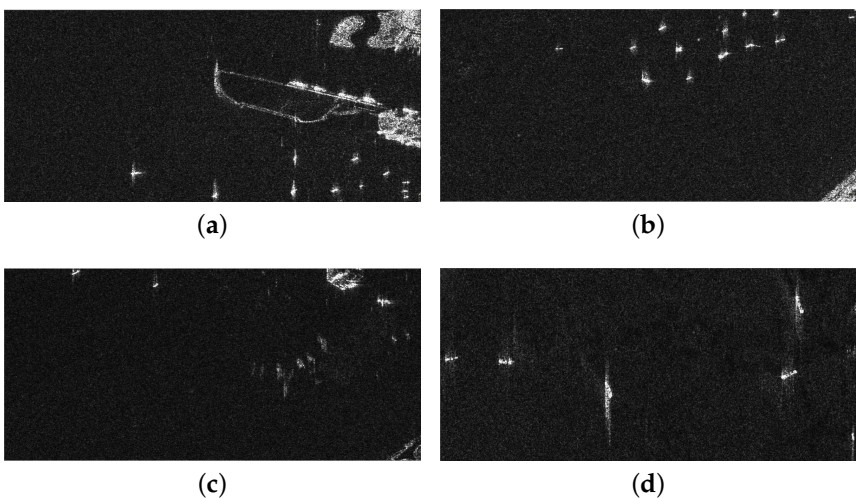

**Figure 9.** Complex dataset ship slices: (**a**,**c**) inshore ship slices; (**b**,**d**) offshore ship slices.

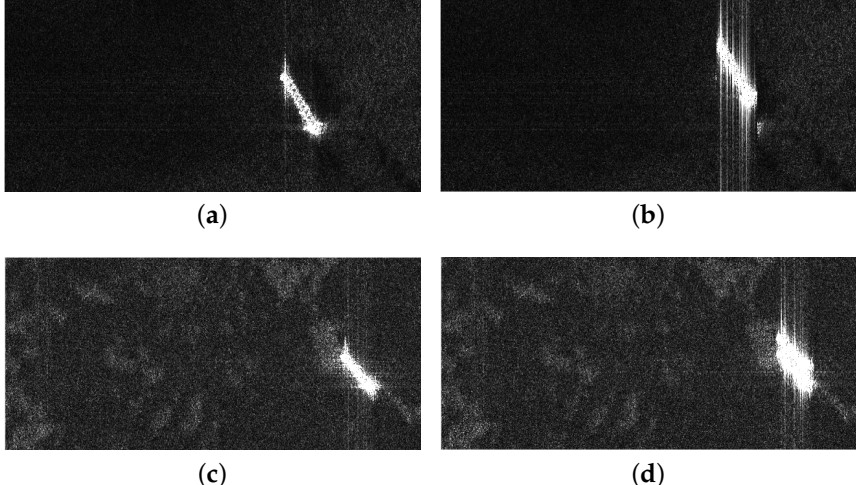

**Figure 10.** Defocused ship slices: (**a**,**c**) normal ship slices; (**b**,**d**) defocused ship slices.

After obtaining the defocused ship slices, the doppler matrix describing the motion state of the target can be calculated by estimating the offset of the doppler center frequency and modulation frequency rate. It can be used as a feature, which helps improve the model's ability to identify defocused ships.

### 3.2. Dataset Description and Settings

The complex ship dataset contains 943 SAR images of 1724 ships with an average image size of 512 × 1024 from the Gaofen-3 satellite, and, on average, each image contained 1.83 ship targets. SAR ships in this dataset are provided with multiple resolutions from 3 m to 10 m and multiple polarizations (VV, VH, HH, HV). We set the ratio of the training set to the test set to 9:1.

Our experiments are carried out under the framework of PyTorch1.1.0, and network training is carried out on computers using Ubuntu16.04 and Cuda10.2. Due to the limitation of the GPU's computing capacity, the maximum number of iterations (epoch) is 12 with a batch size of 4. In addition, ResNet50 pre-trained on ImageNet is adopted as the initialization model. The optimizer uses a stochastic gradient descent (SGD) with a learning rate of 0.25, a momentum of 0.9, and a weight decay of 0.0001. The learning rate drops 10 times at the 8th and 10th epochs to make the model converge. The distribution of Height-width ratios in the dataset is shown in Figure 11.

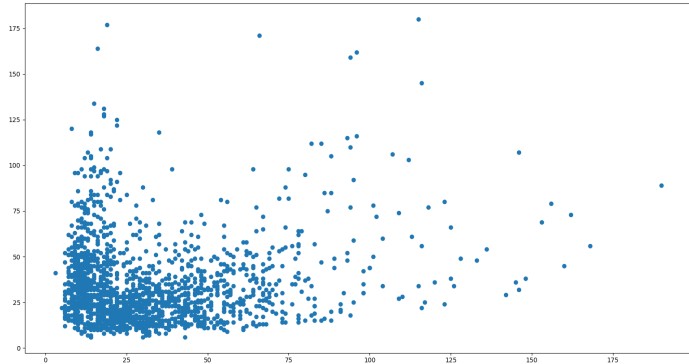

**Figure 11.** Height–width distribution of the bounding box.

*3.3. Evaluation Metrics*

Several metrics are used to evaluate the performance of different ship detection models on the dataset, including recall rate (R), precision rate (P), F1-score, and mean average precision (mAP):

$$Recall = \frac{TP}{TP + FN} \qquad (19)$$

$$Precision = \frac{TP}{TP + FP} \qquad (20)$$

$$F1 - score = 2 \times \frac{Precision \cdot Recall}{Precision + Recall} \qquad (21)$$

$$mAP = \int_0^1 P(R)dR \qquad (22)$$

where TP denotes the number of positive samples that are correctly identified, FP denotes the number of false positive negative samples, and FN denotes the number of false positive samples. In this paper, we use the mAP and F1-score as final measures because they take into account both precision and recall.

*3.4. Ablation Study*

In this section, we verify the designed network model and the effect of fusing doppler features on the results. We conduct ablation experiments on different modules, and the results obtained  are shown in Table 1. It can be seen that due to the influence of the complex background near the coast and the ship image quality, the precision rate of FPN can only reach 86.3%, which will cause a high false alarm rate and further increase the working costs of the sea surface monitoring task. Strengthening the fusion of features and feature map information through the attention mechanism can significantly improve detection performance. As shown in Table 1, after adding the bottom-up feature pyramid, CA, and EA modules, the precision and mAP are improved by 1.4% and 1.6%, respectively. After adding the doppler feature, the mAP of the model is further improved, reaching 96.8%.

Due to the influence of image quality and the sea surface background, the effect of FPN on complex datasets has a greater decline compared to other ship datasets. The defocused ship slices degrade the ability of the network to distinguish targets during the training process, resulting in false detections of defocused noise areas or islands. After refining the features through the attention module and multi-layer pyramids, the model can better distinguish the ship area. Finally, the addition of the doppler center frequency

and modulated frequency rate shift features provide the model with defocusing reference information so that it can distinguish the motion state of the ship. It further improves the accuracy of and reduces the occurrence of false alarms.

**Table 1.** The results of ablation studies.

| Method | Doppler | Attention | mAP | P | R | F1-Score |
|---|---|---|---|---|---|---|
| FPN | × | × | 0.924 | 0.863 | 0.937 | 0.898 |
| FPN with Attention | × | ✓ | 0.940 | 0.877 | 0.942 | 0.908 |
| FPN with Doppler | ✓ | × | 0.943 | 0.892 | 0.931 | 0.911 |
| D-MFPN | ✓ | ✓ | 0.968 | 0.919 | 0.963 | 0.940 |

The inference speed for different models is shown in Table 2, and FPS represents the number of frames of detected pictures per second. It can be seen that compared with the FPN, the D-MFPN only brings a very small inference burden. This is because the CA and EA modules do not use the global attention calculation, and the lightweight design will not affect the overall speed.

**Table 2.** Detection speeds.

| Method | Times | FPS |
|---|---|---|
| FPN | 0.123 | 8.13 |
| FPN with Attention | 0.128 | 7.81 |
| FPN with Doppler | 0.131 | 7.63 |
| D-MFPN | 0.133 | 7.52 |

Different Range Block Method Factors

In the process of obtaining the doppler domain matrix, we use a block method to slice the spectrum along the distance direction. The specific process can be described as follows: according to the preset block length and the preset overlap ratio, a sliding window is performed on the azimuth spectrum along the range direction, then the azimuth spectrum is divided into different blocks, and the incoherent superposition of each block is performed. It can be expressed as:

$$n = \frac{d}{r \times (1 - overlap)} - 1 \tag{23}$$

In the formula, $d$ represents the distance length, which is 512 in this dataset; overlap refers to the preset overlap ratio, and the value range is (0,1), which is 0.5 in the experiment; $r$ is the preset slice length, and n denotes the number of slice images.

Table 3 shows the ablation study results for preset slice length factor n in the range block module. It can be seen from Table 3 that slices of different lengths of the doppler matrix have a significant impact on the detection accuracy. This is because if the slice length is too short, it will not completely envelop the area where the ship is located, and it is easy to generate continuous outliers. If the slice is too long, the target area will be weakened by sea surface features, and no obvious center frequency shift feature will be obtained. Finally, in the D-MFPN, in order to obtain the best detection accuracy, r is set to the best value of 64.

**Table 3.** Different preset slice length factor.

| n | r | Overlap | P | R | mAP | F1-Score |
|---|---|---|---|---|---|---|
| 127 | 8 | 0.5 | 0.874 | 0.923 | 0.928 | 0.897 |
| 63 | 16 | 0.5 | 0.885 | 0.937 | 0.932 | 0.910 |
| 31 | 32 | 0.5 | 0.893 | 0.912 | 0.937 | 0.902 |
| 15 | 64 | 0.5 | 0.919 | 0.963 | 0.968 | 0.940 |
| 7 | 128 | 0.5 | 0.872 | 0.963 | 0.915 | 0.915 |

*3.5. Results on the Dataset*

As shown in Table 4, we provide statistical comparison effects of the D-MFPN with two baseline models. It can be seen that in the moving ship dataset, the D-MFPN's map, recall, and precision significantly outperform the other two algorithms. It is thus proven that the M-DFPN has a good effect on improving the detection performance of ship movement scenarios.

**Table 4.** Comparison results of the D-MFPN and baseline models.

| Method | mAP | P | R | F1-Score |
|---|---|---|---|---|
| Faster RCNN | 0.873 | 0.834 | 0.928 | 0.878 |
| FPN | 0.924 | 0.863 | 0.937 | 0.898 |
| Proposed method | 0.968 | 0.919 | 0.963 | 0.940 |

**4. Discussion**

The detection performance of the D-MFPN is compared with five algorithms, such as the FPN, Cascade RCNN, DCN, PAFPN, and Guide Anchoring. The visualization results of various detection models in offshore and onshore scenarios are shown in Figures 12 and 13, and the index results on the dataset are shown in Table 5.

As can be seen in Figure 12, after adding a large number of defocused ship slices with insignificant geometric features, other detection models become unusually sensitive to ship-like noise and island regions. In Figure 12b–e, the other four algorithms mistakenly detect the noise and island parts in the background as targets, and the overall network has a low degree of discrimination for the ship area. The loss of accuracy caused numerous false alarms, while on the other hand, the D-MFPN successfully detected almost all pictures, even in the case of strong noise interference or low signal-to-noise ratio, which shows that the D-MFPN is very suitable for the ship area and has sharp judgment and robust noise reduction performance. In Figure 12f, compared with other methods, the D-MFPN strengthens the feature structure and can effectively distinguish dense ship features. After adding the doppler domain features, an enhanced judgment ability of defocused pixel blocks is obtained that is less sensitive to defocusing-like areas. Based on the above, the D-MFPN achieves the best performance among all algorithms in the marine background.

The detection results of various methods in nearshore scenarios are shown in Figure 13. Figure 13a shows the corresponding ground truth. As scenarios become more complex, the number of false positives and false positives from other detection algorithms increases. Due to the similarity between ports, land buildings, and ship targets, the redundant box problem becomes more obvious, which further restricts the improvement of model precision. As an important indicator to measure the SAR ship detection algorithm, the false alarm rate may directly affect the actual effect of the detection algorithm. In this scenario, the D-MFPN can still distinguish between complex backgrounds and objects with  excellent performance. As can be seen from Figure 13d, when the ship is in the nearshore scene, there are only a few redundant detection boxes. This is because, for the different blocks along the distance, the doppler domain matrix can effectively distinguish the area where the ship is located, and, for the ships and background pixels in the same block, the attention module and the multi-layer pyramid refinement feature also provide a guarantee of the correct distinction

of ships. It can be proven that the proposed method has good performance in both sea and land scenarios.

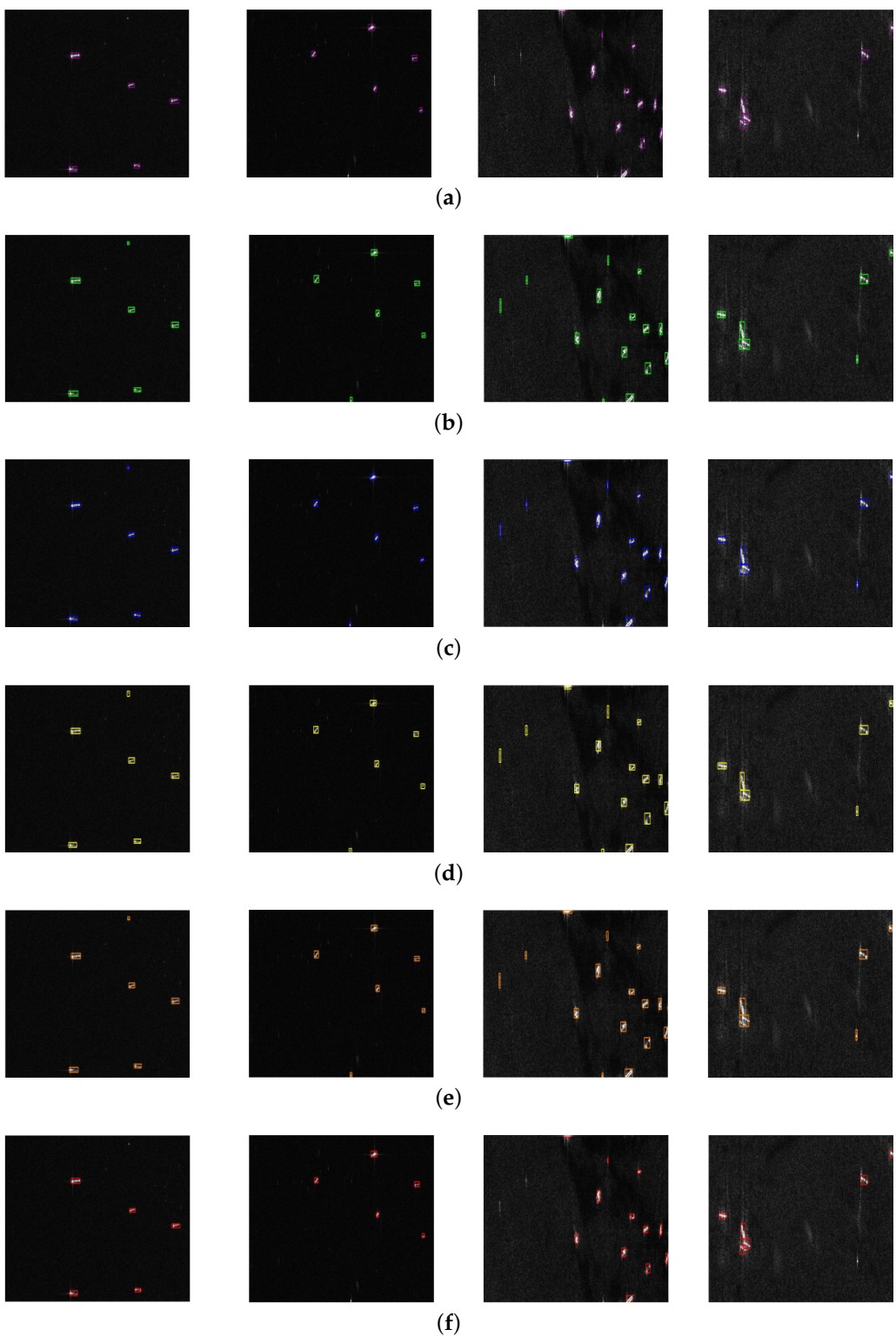

**Figure 12.** Detection results of different methods on offshore images. (**a**) Ground truth; (**b**) Cascade RCNN; (**c**) DCN; (**d**) Guide anchoring; (**e**) PAFPN; (**f**) D-MFPN.

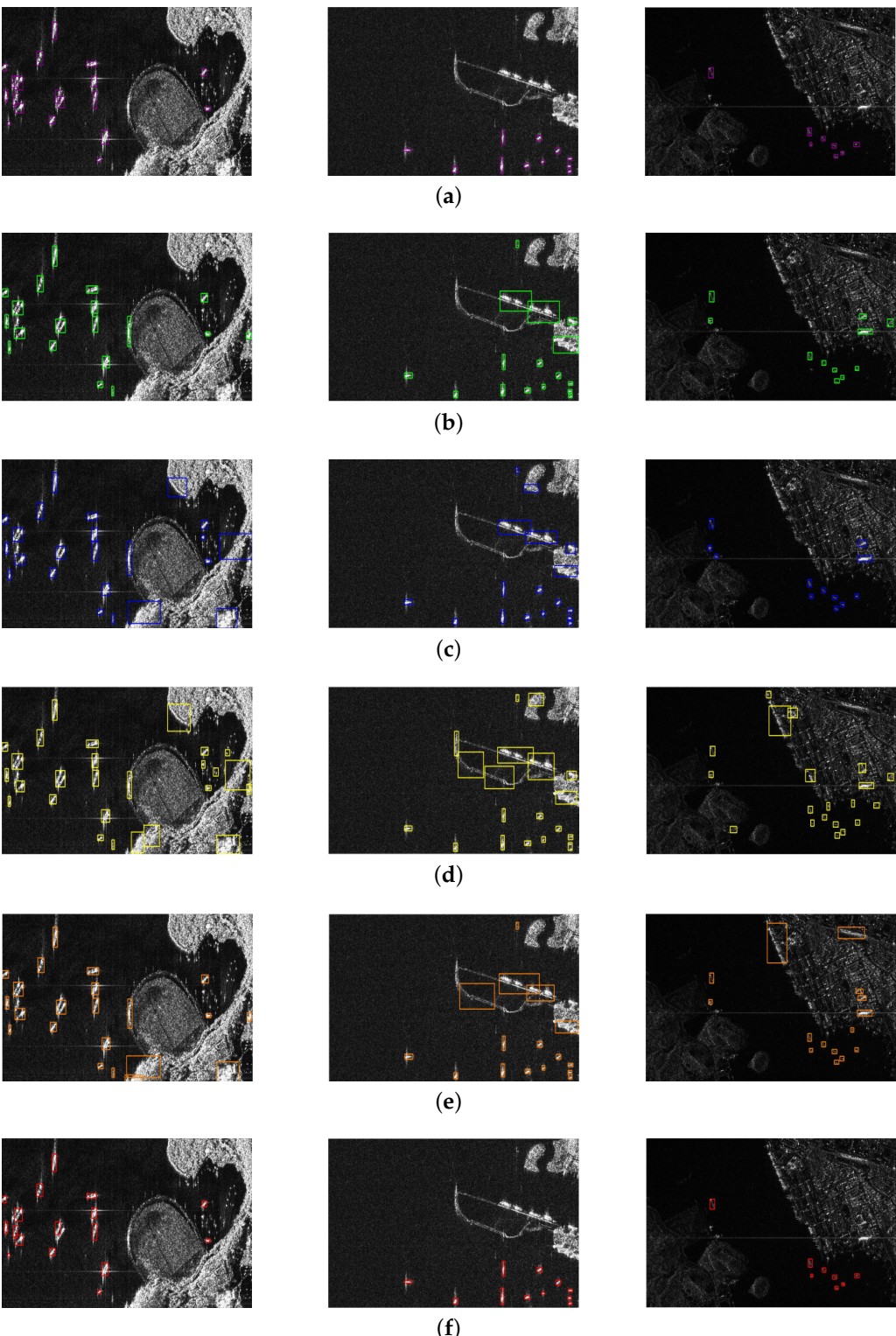

**Figure 13.** Detection results of different methods on inshore images. (**a**) Ground truth; (**b**) Cascade RCNN; (**c**) DCN; (**d**) Guide anchoring; (**e**) PAFPN; (**f**) D-MFPN.

The comparison results of different models are shown in Table 5. The D-MFPN achieves the highest detection accuracy on the dataset, with huge advantages. In particular, the mAP reaches 96.8%, which is 3.2% higher than the second-best CASCADE RCNN. The recall rate reaches 96.3%, which is also the best indicator among all the models. As the network gradually comes to understand the doppler features and accurately expresses

amplitude features during the training process, the D-MFPN not only highlights the features of the target ship area but also suppresses noise and other influences. The proposed method also has good discrimination ability for similar defocused areas. Thus, robust detection performance for moving ships in SAR images is obtained.

**Table 5.** Comparison results of the other state-of-the-art CNN-based methods.

| Method | mAP | P | R | F1-Score |
|---|---|---|---|---|
| Faster RCNN | 0.873 | 0.834 | 0.928 | 0.878 |
| FPN | 0.924 | 0.863 | 0.937 | 0.898 |
| DCN [37] | 0.907 | 0.822 | 0.954 | 0.883 |
| GUIDE ANCHORING [38] | 0.884 | 0.798 | 0.914 | 0.852 |
| CASCADE RCNN | 0.936 | 0.823 | 0.958 | 0.885 |
| PAFPN | 0.905 | 0.785 | 0.955 | 0.862 |
| Proposed method | 0.968 | 0.919 | 0.963 | 0.940 |

## 5. Conclusions

Aiming to address the defocusing phenomenon caused by moving ships, this paper proposes a novel doppler feature matrix fused with a multi-layer feature pyramid network for SAR ship detection. The D-MFPN consists of two branches in the magnitude image. We design an additional bottom-up branch to transfer the underlying location information and combine the CA and EA modules to enhance features and reduce background interference, respectively. In another branch, the doppler feature matrix describing the ship's motion state is obtained by estimating the doppler center frequency and modulation frequency rate offset, which helps to improve the model's ability to distinguish between the foreground and background. After passing through the feature fusion module, the data are sent to prediction. In the experimental section, this paper conducts a detailed ablation study to verify the effectiveness of the two branches. The experimental results on the first proposed Gaofen-3 complex ship dataset show that compared with the other five detection models, the D-MFPN has the best detection ability for moving ships. It also greatly suppresses the generation of false alarms.

Our future work is as follows:

1. We will consider how to add more learnable parameters to the doppler branch in the future.
2. In the future, we will consider continuing to improve the network structure.

**Author Contributions:** Conceptualization, Y.Z.; methodology , Y.Z.; software, Y.Z.; validation, Y.Z., K.F.; formal analysis, Y.Z., K.F.; investigation, Y.Z.; resources, Y.Z.; data curation, Y.Z.; writing and original draft preparation, Y.Z., K.F. and B.H.; writing—review and editing, Y.Z., K.F., B.H., J.Y., Z.P., Y.H. and D.Y.; visualization, Y.Z.; supervision, B.H., K.F.; project administration, B.H.; funding acquisition, B.H. All authors have read and agreed to the published version of the manuscript.

**Funding:** This research was funded by the National Natural Science Foundation of China under Grant Number 41976169.

**Data Availability Statement:** The data of the Gaofen-3 satellite ship slices are provided by the internal data sharing center of the Institute of Aerospace Information of the Chinese Academy of Sciences through the website. For policy reasons, the data on this website cannot be fully disclosed.

**Conflicts of Interest:** The authors declare no conflict of interest.

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
