# Peer review of "D-MFPN: A Doppler Feature Matrix Fused with a Multilayer Feature Pyramid Network for SAR Ship Detection"

_remotesensing, doi:10.3390/rs15030626_

Round 1
Reviewer 1 Report
The paper presents an approach on some relevance for the readers, for enhancing the detection of ships in SAR images, in particular for what concerns moving ships.
The language, clarity and organisation of the paper are on the other hand poor and need significant rework before being able of considering the paper ready for publication.
Several sentences are not complete / the verb is missing or totally unclear, including:
Page 1 Line 31: “Such as CFAR[10,11] and template matching based methods[12,13].”
Page 2 Line 70: “Compared to ordinary optical images.”
Page 2 Line 72: “and the inuity of complex information restricts upper limit of model.”
Page 2 Line 80: “a method to describe SAR ships between different polarization channels.”
Page 8: “Where T represents the target point.”
and several others.
Page 3 Lines 121 and following speak of a Section II and III, but this does not correspond to the real sequence of sections, that are 2 (containing the explanation of the method, and not described in these lines), 3.1 (with the dataset description as should be in II) and following (with experimental results as should be in III).
The paper often uses “defocus” as adjective, what should be replaced by “defocused” or “defucusing”, depending on the circumstances.
The paper presents the algorithm as fully based on the exploitation of the complex pixel values of an SLC image, while it seems more appropriate to mention that this applies only to the “Doppler branch” of the method, while the “Image branch” works on detected (float) backscatter values.
It is not very clear how the results of the Doppler branch (i.e., an estimate of the target radial and azimuthal velocities / Doppler rate) are combined with those of the Image branch, if for example residual focussing is performed to improve the detection capabilities.
The algorithm seems to be based on some blocking / subsetting approach, but it is not clear if / how this is performed, with overlaps etc.
The dataset is obtained from Gaofen-3 data: it is not clear how individual vessels have been identified in the original SAR images. It seems that the effects of motion of the vessels are not due to real motion, but only to simulated defocusing. How can the authors be sure that the identified vessels (before the synthetic defocusing) are not moving? Why real signatures of moving vessels have not been exploited? These is no warranty that the method, tested on simulated data, achieve the same performances on real data. Identification of moving ships can be significantly supported by exploiting AIS data, helping to build a dataset that could avoid the need of simulation.
Even though one of the main advantages of the presented approach should be its robustness / accuracy in case of moving vessels, there is no evaluation of the accuracy of the results in dependence of the velocity of the vessels.
For all these reasons, it is recommended to resubmit the paper only after major modifications and updates.
Reviewer 2 Report
The manuscript presents a doppler features fused with multilayer feature pyramid network(D-MFPN) for SAR ship detection. The experimental results on the Gaofen-3 ship complex dataset illustrate D-MFPN’s optimal performance in defocus ship detection task compared with other 6 competitive convolutional neural network (CNN)-based SAR ship detectors. The novelty is new and applicable, very detailed experiments are provided. The reviewer suggests its acceptance with minor edits.
Comments to the Author
1) The use of English in this paper needs to be moderately improved. All grammar and typing errors need to be corrected.
2) In the selected data set, the ship target is small compared with the original image. Is there any idea for such small target in the design of neural network?
3) What is the generalization ability of the network?
4) When using attention mechanisms, has CA been compared with classic attention mechanisms such as SEnet and CBAM to improve the accuracy of the network?
5) Now the one-stage detection method is also developing very fast, considering the accuracy and speed. Have you tried to compare the performance with the one-stage YOLO series algorithm?
Reviewer 3 Report
Review comments on Zhou et al
D-MFPN: A Doppler Features Fused with Multilayer Feature Pyramid Network for SAR Ship Detection
The aim of the study was to describe a new methodology for analysis of SAR images using a doppler features matrix fused with a multilayer feature pyramid network (D-MFPN). The paper describes the methodological changes that the author made to the existing procedures and describe how each modification improves the outcomes of the analysis. The authors also developed a comprehensive database of ship images based on the Gaofen-3 satellite images. They then compared the results of their new method with existing methods, using the database. The results of the study showed that the proposed technique of using D-MFPN, improved the ability to identify ships in SAR images in nearshore and offshore marine environments, with a high accuracy compared with other methods.
Overall, the data presented in the paper achieves the study aims and provides an original contribution to improving our knowledge and ability on procedures for detection of ships in SAR images. The statistical basis for the changes to image analysis techniques were appropriate to address the aims of the study. As such, I consider that a revised paper incorporating the marked comments on the attached PDF and addressing the comments detailed below could be resubmitted for assessment for publication.
Specific comments on the manuscript are detailed below.
Line 31 This sentence should be revised to clarify its meaning - Such as CFAR[10,11] and template matching based methods[12,13].
In the caption to Figure 7 the reference should be to a satellite and not a star.
Given that the author’s first language is not English, I have marked on the attached PDF document, numerous suggested spelling and grammatical changes to improve the quality and flow of the manuscript. The authors should review these changes and have the manuscript reviewed before resubmission.
However, the number of spelling and grammatical errors in the manuscript indicates that the authors have not bothered to undertake a basic spelling and grammar check of the manuscript using the word processing program before submitting to the journal. This should be done prior to resubmitting any manuscript.
There are several issues with the references with inconsistencies between format of different references. In some references, the journal title e.g., Remote Sensing and Remote sensing Letters are abbreviated as Remote Sens. And Remote Sens. Lett. but the journal title is presented in full for the other references. For some references, the first letter of each word in the paper title is capitalised and should be lower case. In other references, only the first word in the title is capitalised. The authors should check previous issues of the journal regarding presentation of references. In some multi-author references and is used to separate the last two names in some references but not in others.

Round 2
